# An Evaluation of the OLM *Cand*ID Real-Time PCR to Aid in the Diagnosis of Invasive Candidiasis When Testing Serum Samples

**DOI:** 10.3390/jof8090935

**Published:** 2022-09-03

**Authors:** Jessica S. Price, Melissa Fallon, Raquel Posso, Matthijs Backx, P. Lewis White

**Affiliations:** Public Health Wales Mycology Reference Laboratory, PHW Microbiology Cardiff, University Hospital of Wales, Heath Park, Cardiff CF14 4XW, UK

**Keywords:** *Candida* PCR, invasive candidiasis, OLM *Cand*ID, *Candida* diagnostics

## Abstract

Background: Treatment for invasive candidiasis (IC) is time-critical, and culture-based tests can limit clinical utility. Nonculture-based methods such as *Candida* PCR represent a promising approach to improving patient management but require further evaluation to understand their optimal role and incorporation into clinical algorithms. This study determined the performance of the commercially available OLM *Cand*ID real-time PCR when testing serum and developed a diagnostic algorithm for IC. Methods: The study comprised a retrospective performance evaluation of the *Cand*ID real-time PCR assay when testing surplus serum (*n* = 83 patients, 38 with IC), followed by a prospective consecutive cohort evaluation (*n* = 103 patients, 24 with IC) post incorporation into routine service. A combined diagnostic algorithm, also including (1-3)-β-D-Glucan testing, was generated. Results: Prospective *Cand*ID testing generated a sensitivity/specificity of 88%/82%, respectively. Specificity was improved (>95%) when both PCR replicates were positive and/or the patient had multiple positive samples. When combining *Cand*ID with (1-3)-β-D-Glucan testing, the probability of IC when both were positive or negative was >69% or <1%, respectively. Conclusions: The *CandI*D provides excellent performance and a rapid time-to-result using methods widely available in generic molecular diagnostic laboratories. By combining nonculture diagnostics, it may be possible to accurately confirm or exclude IC.

## 1. Introduction

Treatment for invasive candidiasis (IC) is time-critical, with delayed treatment associated with higher hospital mortality [1]. Culture-based tests lack sensitivity and may have a prolonged time to positivity, which may contribute to missed or delayed diagnoses, respectively [2]. Advances in the diagnosis of IC currently lag behind other invasive fungal diseases (IFDs, e.g., invasive aspergillosis or *Pneumocystosis*), in which biomarker testing has significantly improved diagnosis [3,4]. Given the prevalence of IC, it is critical that its diagnosis is optimized to include nonculture methods. In this context, nonculture-based methods for the identification of *Candida*, such as DNA detection by PCR, represent a promising approach, with clinical performance sufficient to allow rapid species-level diagnosis, prompting the initiation of species-oriented therapy soon after the onset of sepsis [5,6]. The combination of mycological tests to provide an optimal diagnosis of IC remains to be confirmed, with the A-STOP clinical trial (ISRCTN43895480) attempting to identify a preferred strategy. While a meta-analysis has generated excellent performance for *Candida* PCR for the diagnosis of candidemia, data are over a decade old, and PCR performance for the detection of other IC manifestations, particularly in the absence of candidemia (e.g., intra-abdominal candidiasis or *Candida* peritonitis) is less robust [6]. This is hindered by the lack of a preferred sample type for molecular detection of IC when the organism is absent from the circulation [2,7]. The availability of commercially manufactured *Candida* PCR platforms helps standardize the process and improves accessibility to such testing. While performance data for automated systems (e.g., T2Candida) is generally encouraging, it remains variable, particularly for sensitivity [8,9]. This could reflect the limitations of targeting an intact organism when it is absent from blood in certain IC manifestations or the influence of antifungal therapy that hypothetically increases the availability of free DNA (DNAemia) [6,7]. Testing cell-free blood fractions (e.g., serum/plasma) targeting circulating DNAemia only (due to prior blood fractionation) may be preferential and improves accessibility to *Candida* PCR testing by employing generic nucleic acid extraction platforms/methods that do not require upstream mycology-specific manipulations (e.g., blood cell lysis and mechanical disruption of the fungal cell), but clinical validation of commercial assays is lacking [7,10].

As part of routine local testing, patients considered at high risk of IC (e.g., abdominal surgery/perforation, intensive care patients (including those admitted with COVID-19), patients with interventions/risks increasing the possibility of IC (venous catheters, broad-spectrum antibacterial agents, total parenteral nutrition, *Candida* colonization)) were prospectively screened for IC by Bruker Fungiplex Candida PCR and Fungitell (1-3)-β-D-Glucan (BDG). As *Cand*ID (OLM Diagnostics, Newcastle-upon-Tyne, UK) has the capacity to increase the level of identification, it was decided to perform a retrospective performance evaluation to determine whether the routine clinical service could be enhanced through the implementation of *Cand*ID. This manuscript documents the process of validating a fungal diagnostic test and its subsequent implementation in a busy routine testing laboratory. It describes the retrospective evaluation of the assay when testing serum, the issues encountered in the transfer to routine diagnostic use, the steps taken to overcome these issues, and the subsequent prospective evaluation and integration into a diagnostic pathway.

## 2. Materials and Methods

### 2.1. Study Design

The study comprised two evaluations. A retrospective, anonymous performance evaluation of the *Cand*ID real-time PCR assay when testing surplus serum, previously tested by Bruker Fungiplex Candida PCR and BDG as a part of routine diagnostic investigations in patients at risk of IC. Initially, serum samples were selected based on a previous positive Fungiplex result, with additional *Cand*ID testing performed on randomly selected samples. This retrospective case/control study was followed by a prospective consecutive cohort evaluation of *Cand*ID real-time PCR assay post incorporation into routine service in place of the Bruker Fungiplex Candida PCR.

Case definition was performed blinded to the *Cand*ID result, with proven IC documented by the recovery of *Candida* sp. from blood or sterile sites. Probable IC was defined in patients with a clinical risk factor for developing IC supported by evidence of *Candida* colonization at ≥2 noncontiguous anatomical sites and a positive serum BDG (with the exclusion of other fungal infections) or recovery of *Candida* from a central venous catheter or deep wound and a positive serum BDG (with the exclusion of other fungal infections). Possible IC was defined in patients with a clinical risk factor for developing IC supported by evidence of *Candida* colonization at ≥2 noncontiguous anatomical sites or a positive serum BDG (with the exclusion of other fungal infections) or recovery of *Candida* from a central venous catheter or deep wound. Justification for the nonproven classifications was based on the potential prevalence of IC associated with various clinical conditions and the subsequently increased probability of IC associated with supporting mycological evidence [11]. For example, in a patient with small bowel perforation and a positive serum BDG, the probability of IC is 39%, which would be increased further by the presence of *Candida* colonization, particularly of central venous catheters or deep wounds [11].

Data obtained to aid in the clinical interpretation of the routine diagnostic PCR results were retrospectively collated as an anonymous performance evaluation with no impact on patient management. Following prior discussions with the local research board, this type of study is not considered research under UK National Health Service guidance and subsequently does not require ethical approval.

### 2.2. Nucleic Acid Extraction

For the retrospective evaluation, total nucleic acid was extracted from 500 µL of serum on the BioMerieux eMAG extraction platform version 1.0.2, using the Generic_3.0.4 extraction protocol with the nucleic acid eluted in 75 µL. Initially, this platform was also utilized for the prospective evaluation, but concerns over contamination (detailed below) resulted in further extractions being performed on the Roche MagNA Pure 96 Extraction Platform. Again, 500 µL of serum was extracted using the MagNA Pure 96 DNA and Viral NA Large Volume extraction procedure, with the nucleic acid eluted in 100 µL. For all extractions, a positive extraction control containing each of the *Candida* species targeted by the *Cand*ID assay (Figure 1) was included together with a negative control.

### 2.3. CandID Real-Time PCR Amplification

Real-time PCR amplification was performed in duplicate following the manufacturer’s instructions using 6 µL of DNA extract in a final volume of 20 µL. For every sample, both the *Cand*ID (targeting *C. albicans*, *C. glabrata*, *C. parapsilosis*, and an internal control) and *Cand*ID PLUS (targeting *C. tropicalis*, *C. krusei*, *C. dubliniensis*, and an internal control) assays were performed as separate reactions within the same run (Figure 1). PCR amplification was performed using the Qiagen Roto-Gene 6000 HRM with the following amplification protocol: Enzyme activation for 2 minutes at 95 °C, followed by 45 cycles of 95 °C for 5 s and 60 °C for 20 s with data acquisition across four channels (green, *C. albicans/C. tropicalis*; yellow, *C. glabrata*/*C. krusei*; red, *C. parapsilosis*/*C. dubliniensis*; and orange, internal control). The internal control target was added to the master mix to identify PCR inhibition only and did not provide evidence on individual sample extraction efficiency. Along with the independent positive and negative extraction controls, a manufacturer-supplied positive PCR control and a no-template PCR control were included in every PCR run. The limit of detection for each target was eight femtograms per PCR reaction (personal communication: Gemma Johnson). When designing the oligonucleotides, a minimum of 40 strains of each target species were aligned to identify sequences with 100% homology, and 2–3 strains of each target species were then physically tested using the optimized assay (personal communication: Gemma Johnson).

### 2.4. (1-3)-β-D-Glucan Testing

BDG testing was performed using the Fungitell Assay (Associates of Cape Cod), testing 5 µL of serum in duplicate according to the manufacturer’s instructions, with a positivity threshold of 80 pg/mL. Samples with a BDG concentration of between 60 and 79 pg/mL were considered indeterminate, and samples below 60 pg/mL were considered negative.

### 2.5. Statistical Analysis

To determine the clinical accuracy of the *Cand*ID assay, the positivity rate in samples originating from cases was compared with the false positivity rate in control samples. To determine clinical performance (sensitivity, specificity, positive and negative likelihood ratios, and diagnostic odds ratio), 2 × 2 tables were constructed using both proven/probable IC and proven/probable/possible IC as true cases and patients with no evidence of fungal disease used as the control population. Performance of the *Cand*ID assay was further assessed by requiring both PCR replicates to be positive or a patient to have multiple positive samples compared with absolute positivity, in which a nonreproducible or a single positive sample were considered significant. Given the case–control study design, predictive values were not calculated for the retrospective study but were generated for the prospective cohort evaluation. For each proportionate value, ninety-five percent confidence intervals and, when required, *p* values (Fisher’s exact test; *p* ≤ 0.05 considered significant) were generated to determine the significance of the difference between rates. To determine an optimal quantification cycle (Cq) threshold for defining *Cand*ID positivity for each of the six *Candida* species targeted by the assay, receiver operator characteristic (ROC) curve analysis was performed. Classification and regression tree (CART) analysis was performed to develop a combined predictive algorithm for IC involving BDG and *Cand*ID testing. Statistical analyses were performed using Graphpad Prism 5 (Graphpad Software, La Jolla, CA, USA) and Microsoft Excel 2016.

## 3. Results

### 3.1. Retrospective Performance Evaluation

A total of 97 serum samples from 83 patients were retrospectively tested by the *Cand*ID assay. Median age of the patient was 58 years (range <1 to 93) with a male/female ratio of 1.96/1, including 19 patients with gastrointestinal/abdominal issues, 18 hematology patients, 10 patients with COVID-19 infection, and 8 patients requiring critical care management for trauma, and a range of patients with other underlying clinical risks (renal (*n* = 5), solid cancer (*n* = 4), surgery (*n* = 4), diabetes (*n* = 3), respiratory (*n* = 3), and other conditions (*n* = 9)). A total of 14 samples originated from 12 cases of proven IC (10 cases of candidemia, 2 cases of *Candida* peritonitis), 16 samples from 12 cases of probable IC, 20 samples from 14 cases of possible IC, and 47 samples from 45 control patients. Positivity rates for samples originating from proven, probable, possible, and no IC were 64% (95% CI: 39–84), 75% (95% CI: 51–90), 55% (95% CI: 34–74), and 6% (95% CI: 2–17), respectively, with false positivity rates in patients with no IC significantly lower than true positivity rates associated with cases of IC, irrespective of the certainty of diagnosis (*p* < 0.0001). The retrospective performance of the *Cand*ID assay when testing serum is shown in Table 1.

Assay specificity appeared optimal, and there were only three false positive cases associated with two *C. krusei* and one *C. parapsilosis* erroneous result. All three false positive results were only positive in one of two replicates, and retrospective specificity could be increased to 100% by requiring both replicates to be positive, but this reduced the sensitivity for detecting proven/probable IC from 79% to 58% and from 66% to 42% for detecting proven/probable/possible IC. ROC analysis identified a threshold of <37 cycles for eliminating false positivity caused by *C. parapsilosis* and *C. krusei*, without significantly impacting sensitivity, albeit IC cases definitely caused by these species were limited. For cases of proven IC with *Candida* isolated from a sterile site that were also positive by the *Cand*ID assay (*n* = 8), concordance between culture identification and molecular identification was 100% (*C. albicans* (*n* = 5), *C. glabrata*, *C. tropicalis*, and *C. glabrata/krusei* (all *n* = 1)).

### 3.2. Prospective Performance Evaluation Using Nucleic acid Extracted Using the BioMerieux eMag Extractor

On the basis of retrospective clinical performance (Table 1), it was decided to incorporate the *Cand*ID assay into routine service. In total, 120 serum samples from eight cases of proven IC (14 samples), five cases of probable IC (9 samples), five cases of possible IC (9 samples), one case of chronic IC (2 samples), and seventy-six patients (86 samples) with no evidence of IC were tested. Sensitivity for the detection of proven/probable and proven/probable/possible IC was similar to retrospective testing at 77% (95% CI: 50–92) and 72% (95% CI: 49–88), respectively. However, the specificity was significantly reduced (75%, 95% CI: 64–83, difference with retrospective specificity: 18%, 95% CI: 4–30, *p* = 0.0139). Environmental monitoring and investigation of extraction and amplification negative controls indicated the source of false positivity was associated with the nucleic acid extraction platform, where negative extraction controls and swabs from the eMAG platform were consistently positive for *C. parapsilosis,* which accounted for 74% (95% CI: 51–88) of clinical false positivity.

### 3.3. Prospective Performance Evaluation Using Nucleic acid Extracted Using the Roche MagNA Pure 96 Extractor

To maintain service, a rapid technical evaluation of the Roche MagNA Pure 96 extractor was performed to exclude contamination with *Candida*. After confirming satisfactory performance, the MagNA Pure 96 extractor was incorporated into routine service for *Candida* PCR, and a prospective consecutive performance evaluation was conducted. Prospective testing using the MagNA Pure 96 extractor involved 175 serum samples from 103 patients comprising 4 cases of proven IC (10 samples), 1 case of probable IC (2 samples), 18 cases of possible IC (48 samples), 1 case of chronic IC (4 samples), and 79 control patients (111 samples). The median age of the patient was 59 years (range <1 to 92), with a male/female ratio of 1.48/1. Underlying conditions included hematological malignancy (*n* = 31), gastrointestinal/abdominal conditions (*n* = 21), COVID-19 infection (*n* = 12), renal issues (*n* = 9), and a range of other conditions (including cardiac, respiratory, solid cancer, and surgery).

Positivity rates for samples originating from proven, probable, possible, chronic, and no IC were 50% (95% CI: 24–76), 50% (95% CI: 9–90), 54% (95% CI: 40–67), 50% (95% C: 15–85), and 13% (95% CI: 8–20), respectively, with false positivity rates in control patients significantly lower than true positivity rates associated with cases of combined IC (*p* < 0.0001).Prospective testing generated an overall sensitivity and specificity of 88% (95 CI: 65–94) and 82% (95% CI: 72–89), respectively (Table 2). 

*C. parapsilosis* accounted for 50% of false positivity, with only 1/7 false positive *C. parapsilosis* results positive in duplicate. Only 4/14 false positive results were positive in both replicates, so applying a positivity threshold requiring both replicates to be positive increased specificity to 95% (95% CI: 88–98; PPV: 76% (95% CI: 53–90); LR +tive: 10.7) with a sensitivity of 54% (95% CI: 35–72; NPV: 87% (95% CI: 79–93); LR -tive: 0.48). No control patients had multiple (≥2) false positive samples generating a specificity/PPV of 100% (LR +tive > 360), but sensitivity/NPV was reduced to 38% and 83%, respectively. Using a threshold requiring multiple serum samples or both PCR replicates to be positive by *Cand*ID before considering a patient to be *Candida* PCR positive generated a sensitivity of 63% (95% CI: 43–79) and specificity of ≥95%.

The median Cq values associated with false positive and true positive results were 35 (range: 32–41) and 34 (range: 26–37) cycles, respectively, limiting the potential of ROC analysis for identifying an optimal Cq threshold, but Cq values of <32 were only associated with cases of IC and Cq values of >37 cycles were only associated with false positive results. Molecular identification confirmed the mycological culture result in 4/5 cases of proven/probable IC.

## 4. Discussion

This two-phase study presents a retrospective case/control evaluation but also a more clinically relevant prospective cohort evaluation of the OLM *Cand*ID assay when testing easily obtainable serum samples that can be conveniently processed using automated nucleic acid extraction platforms, widely available in generic molecular diagnostic laboratories. While the *Cand*ID assay demonstrated comparable performance when testing DNA extracted on both the Roche MagNA Pure 96 and the BioMerieux eMag platforms, the *Candida* contamination experienced on the latter highlights the need for maintaining quality practices through the use of negative controls during every extraction. While the eMag was identified as the source of contamination, it was unclear how this developed, but the design of the instrument employing large reservoirs of buffers to be dispensed through permanent tubing does lend itself to potential fungal contamination. Fungal DNA contamination of the molecular process is well documented; indeed, the predecessor to the eMag (the easyMag) and other commercially available nucleic acid extraction kits have also been associated with fungal DNA contamination [12,13,14,15]. The use of extraction platforms with dispensable or interchangeable reagents individual to each run limits the opportunity for fungal contamination.

The prospective *Cand*ID evaluation utilizing the MagNA Pure 96 for nucleic acid extraction detected all cases of proven/probable IC, and while three cases of possible IC were negative, the certainty of diagnosis is obviously less, and all patients were on antifungal therapy prior to *Cand*ID testing. Interestingly, there was a trend for the overall prospective sensitivity (88%) to be greater than the retrospective (66%) (difference: 22%, *p* = 0.0765), indicating the potential detrimental impact of storage on DNA, despite freezing. It could also explain the numerical difference in prospective (82%) and retrospective (93%) specificity, in which degradation removed much of the low-grade false positivity prior to retrospective *Cand*ID testing. Indeed, the main reason for incorporating the *Cand*ID over the previous Bruker assay was its improved specificity (93%, 95% CI: 82–98) on retrospective evaluation compared with a Bruker specificity of 62% (95% CI: 48–75) generated during routine service. However, the reduced Bruker specificity could reflect previously unidentified low-grade eMag *Candida* contamination, and the prospective *Cand*ID specificity (75%) when using the eMag extractor was not significantly superior to the Bruker assay (*p* = 0.1539).

Disease manifestation can also impact *Candida* PCR performance when testing whole blood, in which the absence of the organism in the circulation limits the potential for molecular processes targeting the intact yeast [16]. While studies have demonstrated the benefit of testing plasma/serum (targeting DNAemia) over whole blood (targeting the *Candida* blastospore) for the molecular diagnosis of deep-seated IC in the absence of circulating organism, the reverse is potentially true for the diagnosis of candidemia [6,7,17,18]. In the prospective arm of this study, all cases of candidemia were successfully detected by the *Cand*ID assay testing serum, and in the retrospective arm, 80% of candidemias were detected despite the potential for sample degradation of the already low burden (<1 CFU/mL blood) of yeast associated with blood culture positivity [19,20]. However, the two missed candidemia cases were also falsely negative during prospective testing by the original Bruker PCR and BDG, with one patient only having *Candida* recovered in 1/3 blood culture bottles and the other receiving prior antifungal therapy. It is not possible to confidently determine the performance of the *Cand*ID assay for the diagnosis of deep-seated IC, with only three cases of *Candida* peritonitis included, two of which retrospectively tested negative by *Cand*ID, with the final case testing positive during the initial prospective evaluation with high rates of false positivity.

In relation to initiating appropriate antifungal therapy, it is important that, when available, the molecular identification is concordant with that of culture. Across the retrospective and both prospective studies, concordance with culture identification was 95% (18/19, 95% CI: 75–99) for cases of proven IC when *Cand*ID was positive. The one discordant result was associated with a case of candidemia caused by *C. albicans* and *C. guilliermondii*, but a molecular identification of *C. parapsilosis*, interestingly at the time of testing the patient also had a significant burden of an unidentified yeast cultured from a central venous catheter. For the four cases of candidemia in the prospective MagNA Pure study, the authorized positive *Cand*ID result was available, on average, 2.25 days earlier than the blood culture result (range: 0–5 days), highlighting the potential to initiate earlier therapy.

In a recent randomized controlled trial, the use of BDG alone to direct antifungal therapy in patients at risk of IC was not associated with improved survival, highlighting the difficulty in interpreting results and accurately diagnosing IC [21]. Previously, Clancy and Nguyen have demonstrated the impact of incidence/prevalence on predictive values of various diagnostic tests as a useful guide to aid in the interpretation of results in various clinical cohorts [11]. In an attempt to take this further, CART analysis was undertaken to permit the incorporation of both *Candida* PCR and BDG testing into a single algorithm (Figure 2). 

Using BDG as a primary test of 1000 patients with an incidence of IC of 10%, the probability of IC associated with a positive and negative BDG result is 31% and 3%, respectively. While an increasing BDG concentration can increase the likelihood of IFD, false positivity can be associated with high BDG levels (e.g., IVIG), and, obviously, BDG true positivity indicates IFD, not just IC [24]. BDG specificity can be improved through sequential positivity, but this involves additional sampling, potentially delaying diagnosis [25]. Incorporating *Cand*ID testing on the same serum sample potentially improves diagnostic understanding while providing species identification and minimizing delay. In patients already positive by BDG, the probability of IC in a patient who is also *Cand*ID positive in 1/2 replicates on a single occasion is 69%, whereas the probability of IC when the *Cand*ID is negative in a patient who was initially BDG negative is <1%. If a more stringent PCR positivity threshold requiring both PCR replicates to be positive is applied, then the probability of IC in a BDG-positive patient is increased to 85% (Figure 2).

A range of alternative commercially available *Candida* PCR assays is currently available, although other than the T2Candida assay, clinical validation is limited [5,8]. The performance of *Cand*ID is comparable with that of other assays but does provide an increased range of species identification, particularly over the Bruker Fungiplex *Candida*, which has been noted as a limitation of this test [26]. The T2Candida assay has undergone significant clinical evaluation, and while meta-analytical performance is excellent, both sensitivity and specificity can vary, and testing is limited to whole blood, which may not be optimal in the absence of candidemia [8]. While the fully automated T2 approach minimizes interassay variability, it dictates the use of bespoke equipment dedicated to this process and is associated with considerable expense, both of which limit its introduction into already equipped generic molecular diagnostic laboratories. The potential to incorporate novel commercial assays such as the Fungiplex or *Cand*ID onto existing multiuse platforms is beneficial.

## 5. Conclusions

In conclusion, *CandI*D provides excellent performance and a rapid time-to-result with nucleic acid extraction through PCR amplification, and result interpretation was completed in less than 4 h. Species identification is concordant with culture and may prove beneficial in cases of suspected IC when culture is lacking by avoiding treatment delay or inappropriate therapy choices for IC caused by non-*albicans* species. The use of prior antifungal therapy compromises *Cand*ID sensitivity, but a combined PCR/BDG strategy limits missing cases. Specificity can be improved by requiring multiple positive samples, both PCR replicates to be positive, or by combining the positive *Cand*ID result with BDG positivity. In patients at high risk of IC, combining BDG testing with *Cand*ID PCR provides a strategy to both exclude and—even more so—confirm IC in the absence of culture, but larger-scale confirmatory studies are needed.

## Figures and Tables

**Figure 1 jof-08-00935-f001:**
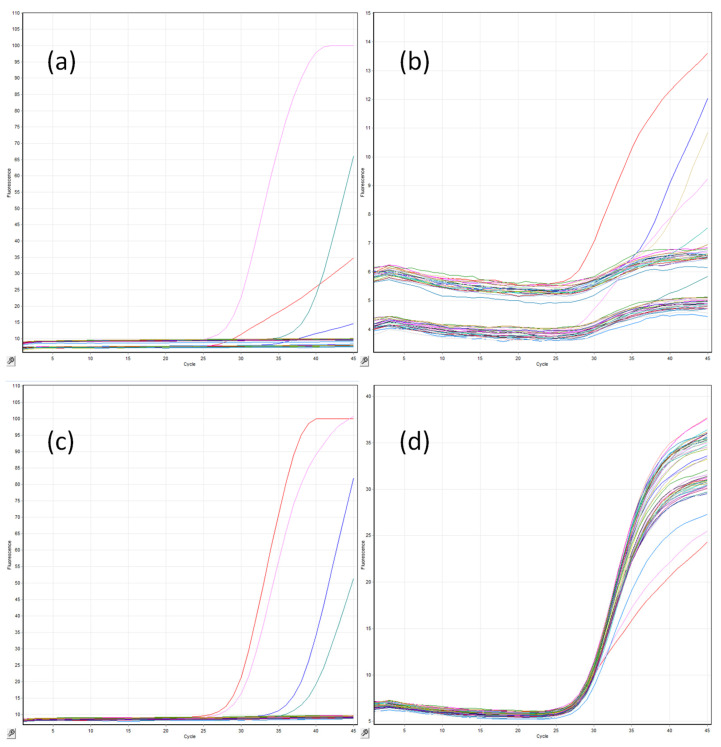
A typical OLM *Cand*ID and *Cand*ID plus multiplex real-time PCR tests on the Qiagen Rotorgene 6000 HRM instrument for the detection of (**a**) *C. albicans*/*C. tropialis*, (**b**) *C. dubliniensis/C. parapsilosis*, (**c**) *C. glabrata/C. krusei*, and (**d**) internal control, in which the horizontal axis reflects the number of cycles (*n* = 45), and the vertical axis is the absolute fluorescence for each channel.

**Figure 2 jof-08-00935-f002:**
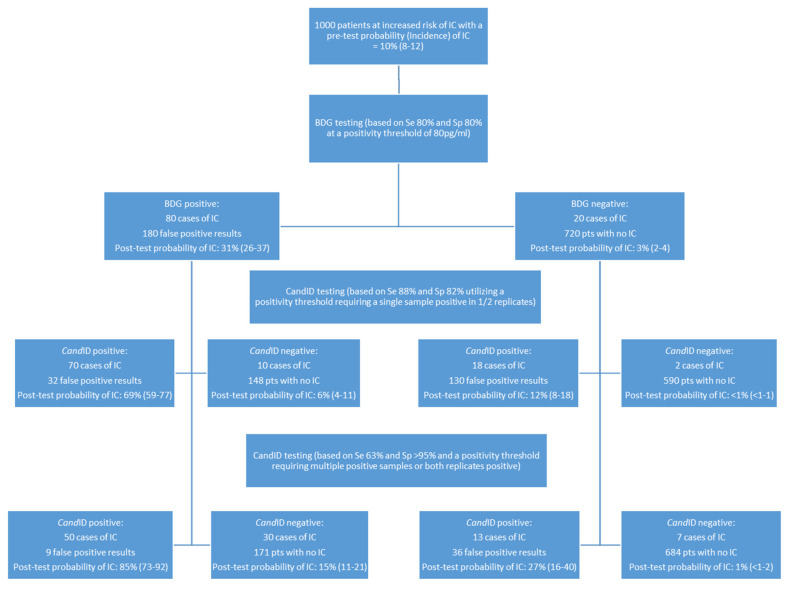
A classification and regression tree (CART) algorithm incorporating Fungitell (1-3)-β-D-Glucan (BDG) and OLM *Cand*ID real-time PCR testing for predicting invasive candidiasis (IC) in patients at increased risk of IC (incidence 10%), based on clinical prediction models for candidemia or patients post emergency surgery for intra-abdominal infection or with colonic perforation, as defined by Clancy and Nguyen [9]. The probability of IC is provided together with the 95% interval in parentheses. Sensitivity and specificity values for BDG testing for the diagnosis of IC are comparable with those generated by systematic review and meta-analysis of BDG testing and are in line with those used in previous predictive studies for IC [11,22,23]. Sensitivity and specificity values for *Cand*ID are those generated in the prospective arm of this current study when testing serum.

**Table 1 jof-08-00935-t001:** Retrospective performance of the OLM *Cand*ID real-time PCR when testing serum in comparison to routine prospective testing using the Bruker Fungiplex Candida Assay.

Population (*n* = 83)	Parameter									
	Se (%, 95% CI)		Sp (%, 95% CI)		LR +tive		LR -tive		DOR	
	CID	BFC	CID	BFC	CID	BFC	CID	BFC	CID	BFC
Candidemia (10) vs. no IC (45)	80 (49–94)	70 (40–89)	93 (82–98)	62 (48–75)	11.4	1.8	0.2	0.5	53.1	3.8
Probable IC (12) vs. no IC (45)	92 (65–99)	75 (47–91)	93 (82–98)	62 (48–75)	13.1	2.0	0.1	0.4	152.8	4.9
Candida peritonitis (2) vs. no IC (45)	0 (0–66)	50 (10–91)	93 (82–98)	62 (48–75)	0	1.3	1.1	0.8	0	1.6
Combined proven/prob IC (24) vs. no IC (45)	79 (60–91)	71 (51–95)	93 (82–98)	62 (48–75)	11.3	1.9	0.2	0.5	50.0	4.0
Possible IC (14) vs. no IC (45)	43 (21–67)	64 (39–84)	93 (82–98)	62 (48–75)	6.1	1.7	0.6	0.6	10.0	2.9
All IC (38) vs. no IC (45)	66 (50–79)	68 (53–81)	93 (82–98)	62 (48–75)	9.4	1.8	0.4	0.5	25.8	3.5

Key: Se, sensitivity; Sp, specificity; LR +tive, positive likelihood ratio; LR -tive, negative likelihood ratio; DOR, diagnostic odds ratio; CID, OLM *Cand*ID; BFC, Bruker Fungiplex Candida; IC, invasive candidiasis.

**Table 2 jof-08-00935-t002:** Prospective performance of the OLM *Cand*ID real-time PCR when testing serum with DNA extracted using the Roche MagNA Pure 96.

Population (*n* = 103)	Parameter						
	Se (%, 95% CI)	Sp (%, 95% CI)	PPV (%, 95% CI)	NPV (%, 95% CI)	LR +tive	LR -tive	DOR
Candidemia (4) vs. no IC (79)	100 (51–100)	82 (72–89)	22 (9–45)	100 (94–100)	5.6	<0.001	>4571
Probable/chronic IC (2) vs. no IC (79)	100 (34–100)	82 (72–89)	13 (4–36)	100 (94–100)	5.6	<0.001	>4571
Combined candidemia/prob/chronic IC (6) vs. no IC (79)	100 (61–100)	82 (72–89)	30 (15–52)	100 (94–100)	5.6	<0.001	>4571
Possible IC (18) vs. no IC (79)	83 (61–94)	82 (72–89)	52 (34–69)	96 (88–98)	4.7	0.20	22.9
All IC (24) vs. no IC (79)	88 (69–96)	82 (72–89)	60 (44–74)	96 (88–98)	4.9	0.15	32.5

Only one case of candidemia was PCR positive in duplicate and/or PCR positive on multiple samples. Only one case of probable/chronic IC was PCR positive in duplicate and/or PCR positive on multiple samples. Thirteen cases of possible IC were PCR positive in duplicate and/or PCR positive on multiple samples. Only four control patients were PCR positive in duplicate. Key: Se, sensitivity; Sp, specificity; PPV, positive predictive value; NPV, negative predictive value; LR + tive, positive likelihood ratio; LR-tive, negative likelihood ratio; DOR, diagnostic odds ratio.

## Data Availability

Not applicable.

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
