# Peer review of "An Evaluation of the OLM CandID Real-Time PCR to Aid in the Diagnosis of Invasive Candidiasis When Testing Serum Samples"

_jof, 2022, doi:10.3390/jof8090935_

Round 1

Reviewer 1 Report

In this manuscript " An evaluation of the OLM CandID real-time PCR to aid in the diagnosis of invasive candidiasis when testing serum samples”, Jessica S. Price., et. al. based on a retrospective, made an effort to determine the performance of the commercially available OLM CandID real-time PCR when testing serum and developed a diagnostic algorithm for invasive candidiasis

The comments and suggestions for this manuscript are as follows-

1.    The introduction and the main body of the manuscript are a typical textbook type. This is lacking intellectual input from authors. The author must provide a comprehensive introduction and simplified discussion with proper references.

2.    On page 2, in the study design section, the author has described “A retrospective, anonymous performance evaluation of the CandID real-time PCR assay when testing surplus serum, previously tested by Bruker Fungiplex Candida PCR and BDG as a part of the routine, diagnostic investigations in patients at risk of IC”. To a better understanding of the readers, the author must provide a comparative table (CandID real-time PCR, Bruker Fungiplex Candida PCR, and BDG) showing sensitivity (Se) and specificity (Sp), in the proven IC, Probable IC, and developing IC conditions.

3.    How many Candida sp. strains can be tested with the OLM CandID real-time PCR diagnostic kit? If possible, the author may provide the PCR primer sequence for each target in the supplementary document.

4.    What is the sensitivity/lower detection limit of this kit, in terms of DNA quantification (nanograms)? In the material method section (page 4), it is reported as 6ul. Can it be used for the early diagnosis of Candida sp.?

5.    Page 4, in the result section, it is mentioned that “A total of 97 serum samples from 83 patients were retrospectively tested by the CandID assay”. Page 6, “Testing involved 175 serum samples from 103 patients”. The author should clarify the statement (mode/method of sample collection).

6.    Page 3, figure 1. The figure needs formatting with proper X and Y-axis legends and better resolution. Likewise page 9, figure 2 needs higher resolution for better visibility.

Author Response

We thank the reviewer for the extensive review and insightful comments which we have attempted to address to the best of our abilities.

Comment 1:  

The introduction and the main body of the manuscript are a typical textbook type. This is lacking intellectual input from authors. The author must provide a comprehensive introduction and simplified discussion with proper references.

Response:

We feel this reflects a difference in writing styles and given this manuscript is a clinical validation the subsequent structure and content should reflect this. Indeed, the other reviewer did not raise any concerns regarding the structure and input of the paper.

We feel the introduction should succinctly provide background to the subject (in this case the Molecular diagnosis of IC) and briefly outline the associated study. We feel the introduction already meets this brief. 

It appears the reviewer wishes us to focus more on the introduction than the discussion, but we feel the discussion which reviews the results and discusses the potential implications of these results should take preference over the introduction for both content and intellectual input. In the discussion we have proposed a novel diagnostic algorithm that we feel represents significant intellectual input and could advance the diagnosis of IC. 

While we feel the current introduction meets the requirements for an introduction to the study, we have made some minor alterations to the text and references accordingly. 

We feel the current materials and methods is adequate, and has not been changed.

In the results section, we have amended the headings and introduced an extra heading in this section to break up the text.

We feel the discussion is fit for purpose, although it has been amended to reflect the comments of reviewers and we have adjusted references.

Comment 2:

On page 2, in the study design section, the author has described “A retrospective, anonymous performance evaluation of the CandID real-time PCR assay when testing surplus serum, previously tested by Bruker Fungiplex Candida PCR and BDG as a part of the routine, diagnostic investigations in patients at risk of IC”. To a better understanding of the readers, the author must provide a comparative table (CandID real-time PCR, Bruker Fungiplex Candida PCR, and BDG) showing sensitivity (Se) and specificity (Sp), in the proven IC, Probable IC, and developing IC conditions.

Response:

Table 1: Now contains the data on the performance of the Bruker Fungiplex Candida.

Comment 3:

How many Candida sp. strains can be tested with the OLM CandID real-time PCR diagnostic kit? If possible, the author may provide the PCR primer sequence for each target in the supplementary document

Response:

The following text has been included: When designing the oligonucleotides a minimum of 40 strains of each target species were aligned to identify sequences with 100% homology and 2-3 strains of each target species were then physically tested using the optimized assay.

Given this is a commercially developed assay it is not possible to disclose the oligonucleotide sequences.

Comment 4:

What is the sensitivity/lower detection limit of this kit, in terms of DNA quantification (nanograms)? In the material method section (page 4), it is reported as 6ul. Can it be used for the early diagnosis of Candida sp.?

Response: The following text has been included: The limit of detection for each target was eight femtograms per reaction.

We feel this is comparable with other real-time PCR assays, and at a level that would permit an early detection of Candida, however,  this would require a larger prospective study. We included the following statement in the discussion: For the four cases of candidaemia in the prospective MagNA Pure study, the authorized positive CandID result was available, on average, 2.25 days earlier than the blood culture result (range: 0-5 days), highlighting the potential to initiate earlier therapy.

Comment 5:

 Page 4, in the result section, it is mentioned that “A total of 97 serum samples from 83 patients were retrospectively tested by the CandID assay”. Page 6, “Testing involved 175 serum samples from 103 patients”. The author should clarify the statement (mode/method of sample collection)

Response:

This reflects the different number of samples tested retrospectively (page 4) and prospectively (page 6). We feel the associated headings already highlight this difference. However we have now highlighted that the latter numbers reflect prospective testing using the MagNA Pure 96 extractor and amended the headings to ease reading.

Comment 6:

Page 3, figure 1. The figure needs formatting with proper X and Y-axis legends and better resolution. Likewise page 9, figure 2 needs higher resolution for better visibility.

Response:

The legend to figure 1 now contains the following text "where the horizontal axis reflects the number of cycles and the vertical axis is the absolute fluorescence for each channel."

In relation to figure 2, we are limited by the current page layout, but would not object to this figure being rotated 90° to allow the figure to be enlarged. We leave this decision to the editorial team.

Reviewer 2 Report

The manuscript “An evaluation of the OLM CandID real-time PCR to aid in the diagnosis of invasive candidiasis when testing serum samples” presents a two-stage evaluation (retrospective case-control and prospective cohort evaluation) of the OLM CandID real-time PCR assay in serum samples. The design of the study is adequate, the presentation and the writing allow a clear understanding of the results obtained, which justifies the authors' conclusion. The authors rightly recommend the use of other tests, such as the BDG test, to support the PCR results and to achieve a better clinical interpretation of the PCR results. It would be worthwhile for the authors to discuss the advantages and disadvantages of the analyzed assay with respect to other real-time PCR assays aimed at supporting the diagnosis of IC.

Author Response

We thank the reviewer for their comprehensive review of the manuscript and thoughtful comment, which we have addressed as follows in discussion:

Comment:

It would be worthwhile for the authors to discuss the advantages and disadvantages of the analyzed assay with respect to other real-time PCR assays aimed at supporting the diagnosis of IC.

Response:

A range of alternative commercially available Candida PCR assays are currently available, although other than the T2Candida assay, clinical validation is limited. [3] Performance of the CandID is comparable with that of other assays, but does provide an increased range of species identification, particularly over the Bruker Fungiplex Candida, which has been noted as a limitation of this test. [22] The T2Candida assay has undergone significant clinical evaluation, and while meta-analytical performance is excellent, both sensitivity and specificity can vary and testing is limited to whole blood, which may not be optimal in the absence of candidaemia. [3] While the fully automated T2 approach minimizes inter-assay variability, it dictates the use of bespoke equipment dedicated to this process and is associated with considerable expense, both of which limit its introduction into already equipped generic molecular diagnostic laboratories. The potential to incorporate novel commercial assays such as the Fungiplex or CandID on to existing multiuse platforms is beneficial.